# *In Silico* Research of New Therapeutics Rotenoids Derivatives against *Leishmania amazonensis* Infection

**DOI:** 10.3390/biology11010133

**Published:** 2022-01-14

**Authors:** Adrián Vicente-Barrueco, Ángel Carlos Román, Trinidad Ruiz-Téllez, Francisco Centeno

**Affiliations:** 1Departamento de Bioquímica y Biología Molecular y Genética, Facultad de Ciencias, Universidad de Extremadura, 06071 Badajoz, Spain; adribarru@gmail.com; 2Departamento de Biología Vegetal, Ecología y Ciencias de la Tierra, Facultad de Ciencias, Universidad de Extremadura, 06071 Badajoz, Spain; truiz@unex.es

**Keywords:** *Leishmania amazonensis*, CADD, ODC, docking, deguelin, protein modeling

## Abstract

**Simple Summary:**

In this work, we show bioinformatic procedures to search for new drugs form plant-derived molecules, previously described in ethnobotany as useful for the treatment of leishmaniasis.

**Abstract:**

Yearly, 1,500,000 cases of leishmaniasis are diagnosed, causing thousands of deaths. To advance in its therapy, we present an interdisciplinary protocol that unifies ethnobotanical knowledge of natural compounds and the latest bioinformatics advances to respond to an orphan disease such as leishmaniasis and specifically the one caused by *Leishmania amazonensis*. The use of ethnobotanical information serves as a basis for the development of new drugs, a field in which computer-aided drug design (CADD) has been a revolution. Taking this information from Amazonian communities, located in the area with a high prevalence of this disease, a protocol has been designed to verify new leads. Moreover, a method has been developed that allows the evaluation of lead molecules, and the improvement of their affinity and specificity against therapeutic targets. Through this approach, deguelin has been identified as a good lead to treat the infection due to its potential as an ornithine decarboxylase (ODC) inhibitor, a key enzyme in *Leishmania* development. Using an in silico-generated combinatorial library followed by docking approaches, we have found deguelin derivatives with better affinity and specificity against ODC than the original compound, suggesting that this approach could be adapted for developing new drugs against leishmaniasis.

## 1. Introduction

Leishmaniasis is a chronic pathology produced by the infection of intracellular protozoa of the genus *Leishmania*, usually transmitted by biological vectors, among which *Phlebotomus* and *Lutzomyia* genera are the most important ones [1]. Currently, it is estimated that there are 12 to 15 million people suffering from leishmaniasis worldwide, with between 1,500,000 and 2,000,000 new infections (350 million at risk) and 70,000 deaths each year [1,2].

WHO classifies leishmaniasis in three main forms: visceral, cutaneous, and mucocutaneous [3]. Among the three types, cutaneous leishmaniasis (CL) is the most prevalent. It is characterized by the generation of ulcerous lesions on the skin that cause permanent scars [3]. The available data from the WHO (2020) [4] report that cutaneous leishmaniasis is widespread throughout the world, being endemic in 90 countries, with the Midwest, Central Asia, the Mediterranean area, and America concentrating 95% of the infections. Of the 20 species that have been reported to affect humans [5], *Leishmania amazonensis* is one of the most relevant species in terms of epidemiology because is endemic in Central and South America [6,7] assuming approximately 8% of cases in some highly affected areas such as Brazil [8,9]. Although *L. amazonensis* is one of the main species responsible for CL [10], it has been observed that its infection can be associated with mucocutaneous leishmaniasis (MCL) and visceral leishmaniasis (VL) [11].

Despite being an endemic disease in many countries, with a high incidence and severity, there are very few effective drugs in the prevention and control of the disease. Although some drugs are available for the treatment of leishmaniasis (Table 1), they present major problems such as high toxicity, limited efficacy, low oral bioavailability, as well as the generation of resistance by the parasites [12,13,14]. Therefore, new drugs with better pharmacological properties are required. Despite the need for the development of new therapies, leishmaniasis is a prevalent pathology being considered as an orphan disease.

In the drug development process, plants have always played an important role. According to WHO data, 11% of medicines are plant-based and 80% of the world’s population relies on traditional medicine based on plant extracts (more than 50,000 plants with therapeutic potential are known) [22]. In this context, ethnobotany studies the human use of plant species. This knowledge is very useful in the field of drug development since it can be used to obtain multiple lead compounds for the pharmaceutical industry [23]. The main advantage of ethnobotanical knowledge is the fact that it is based on trial/error tests carried out over long periods of time. This also has an impact on the evaluation of toxicity (which is one of the most expensive phases in drug development), since ethnobotany offers us the knowledge of the concentrations in which the active principles of plants must be used so that they are not toxic or their toxicity is the least possible [24].

Computer-aided drug design (CADD) has led to an exponential increase in pharmacological studies as they speed up the development process by taking advantage of chemical and biological information. Therefore, lead compounds can be obtained with a very low economic investment compared to traditional approaches [25]. These techniques make it possible to perform studies to determine the effectiveness of different drugs against a target before initiating in vivo trials. Docking is one of the most widely used techniques as it allows to predict the ligand (drug) binding site in the protein, the ligand conformation in this site, as well as the estimation of the energy of the binding process, thus estimating the affinity of the ligand for its target [26]. Although many protein sequences are known, very few protein structures are available, which is a bottleneck for the development of new drugs. However, in recent years, by in silico protein modeling, these structures are now being obtained from the sequences with results very close to those that would be obtained by crystallographic methods [27]. This work aims to combine ethnobotanical knowledge with the latest advances in CADD, something that has already been seen to be successful in *Leishmania* [28,29], to contribute to the development of new drugs for the treatment of *L. amazonensis* infection, as well as to serve as a protocol applicable to other orphan diseases.

## 2. Materials and Methods

### 2.1. Bibliographical Review and Identification of Leads

Since the purpose is to identify a possible treatment against *L. amazonensis*, we propose that inhibiting one of the key biosynthetic pathways for the parasite would lead to the death of the parasite and consequently mitigate the effects of the disease. To this end, the published literature was reviewed in search of possible pharmacological targets for this parasite. Likewise, assessment of anti-protozoal activitiy of plants traditionlly used in Ecuador in the treatment of leishmaniasis was considered [30] and ethnobotanical information from the kiwchwa communities, we had previously worked with [31] was used and shared under the Nagoya Protocol Frame. This is due to the fact that the Kichwas live in the Amazon region, an area in which *Leishmania amazonensis* is mainly distributed [32].

From the identified plants, a bibliographic review of their main metabolites was carried out. With the information on possible targets of interest and metabolites, SwissTargetPrediction [33] (using *Homo sapiens* as a reference specie due to the expected homology between enzymes) was accessed in order to find the probability of a match between compounds and targets. Probabilities higher than 0.85 (0.85 is the threshold for the Manhattan-based similarity on 3D [33]) were considered significant, allowing us to identify a possible lead molecule. Once a potential lead and its target were identified, the literature was reviewed for potential side effects and the pharmacological targets involved.

### 2.2. Lead Compound Improvement and Enhancement

After identifying therapeutic targets and off-targets, the objective was to improve therapeutic action and minimize side effects. For this purpose, a combinatorial library was generated and tested by docking. We first resort to the search for protein structures resolved by X-ray diffraction or Cryo-EM (RSCB PDB) [34] structures. Alternatively, if no crystallized structure was available, an in silico folding by protein modeling was performed, based on the previously validated genetic information available in databases. For this purpose, I-Tasser [35,36,37], locPremdf [38], and QMeanDisCO3 [39] were used.

The library of molecules was generated by SmiLib v2.0 [40] using the most common substituents and spacers from MOLINSPIRATION [41]. 3D SDF file was created using CACTVS (https://cactus.nci.nih.gov/translate/, accessed on 15 May 2021). The free web server DockThor (https://www.dockthor.lncc.br/v2/, accessed on 24 May 2021) [42,43] was used for docking. ADME-TOX properties of the top 5 selected compounds were calculated using Swiss-ADME (http://www.swissadme.ch/, accessed on 29 December 2021). Finally, the docking results obtained for each molecule were represented as the affinity for the therapeutic target protein versus the affinity for the toxic target. The detail of the possible interactions at the binding site with its targets was carried out for the molecule with higher affinity for therapeutic target and lowest affinity for toxic target using Protein-Ligand Interaction Profiler (PLIP) (https://plip-tool.biotec.tu-dresden.de/plip-web/plip/index, accessed on 24 May 2021) [44].

## 3. Results

### 3.1. Selected Target and Query Phytocompound

Using SwissTargetPrediction, matches were obtained between the identified plant metabolites and the protein targets of interest in *Leishmania*. Among the plants used by the *kichwa* communities to treat mosquito bites (vector of leishmaniasis transmission) and animal bites, the following species were described: *Artocarpus altilis*, *Brunfelsia grandiflora subsp. grandiflora*, *Calliandra angustifolia*, *Hyptis obtusiflora*, *Lonchocarpus utilis*, and *Nicotiana tabacum*. The main secondary metabolites of these plants (20 compounds) (Appendix A) were tested in SwissTargetPrediction (Appendix A) to identify their potential targets.

Among the targets of the 20 molecules studied, there was a coincidence with a key enzyme for the parasite. The best combination was obtained for ornithine decarboxylase (ODC) and deguelin (Figure 1), with some studies describing this association [45,46]. Studies have also shown that this family of molecules can also produce undesired side effects and can be considered natural toxins, they are inhibitors of the NADH:ubiquinone oxidoreductase (NUO) [46,47].

### 3.2. Pharmacological Optimization of Deguelin

#### 3.2.1. Three-Dimensional Structures

The structure of the human NUO is available in the RCSB PDB (5XTD) [48], but the ODC of *L. amazonensis* is not. After an extensive search in genomic databases, the only information available was a partial mRNA (EU429358.1—*Leishmania amazonensis* clone AR205 ornithine decarboxylase (ODC)-like mRNA, partial sequence) from GenBank [49], so a validation was necessary to determine if it corresponded to *L. amazonensis* ODC.

#### 3.2.2. Assessment of ORF Sequences for *L. amazonensis* ODC

First, a multiple alignment was performed in M-Coffee (http://tcoffee.crg.cat/apps/tcoffee/do:mcoffee, accessed on 24 October 2018) [50] (Appendix A) with reviewed mRNAs from other *Leishmania* species, showing high similarity. After that, the presence of open reading frames (ORF) was checked with NCBI ORFfinder (https://www.ncbi.nlm.nih.gov/orffinder/, accessed on 15 April 2021), with an ORF1 from the start to the end of the sequence (Appendix A). The aminoacid sequence obtained from ORF1 (Appendix A) was submitted to Uniprot Blast (https://www.uniprot.org/blast/ accessed on 25 October 2018) obtaining as a result that this sequence presents high similarity with ODC from other species of the *Leishmania* genus (Figure 2). In addition, a tblastn analysis was performed against the *L. amazonensis* genome database (http://bioinfo08.ibi.unicamp.br/leishmania/ accessed on 12 May 2019) [51] (Appendix A), obtaining some gaps. These results suggest that mRNA EU429358.1 may, in fact, be a complete and processed mRNA of *L. amazonensis* (Figure 2).

The ORF1 sequence for *L. amazonensis* ODC was then modeled with I-Tasser and iteratively refined 20 times with locPremdf (Appendix A) to obtain the most reliable three-dimensional model possible. To validate the improvement, QMeanDisCo was used as an external validator, and the model resulting from refinement 10 was found to be the best among the 20 performed (Figure 3) (Appendix A).

This model presents better folding parameters and resolution compared to the original structure (Figure 4). In the left panel, we can see that the refined structure shows a bluer tone (better resolution), in the central panel of the graph we can see how the QMeanDisCo score is higher, especially at the N-terminal end. These may be due to the presence of a dimerization domain present in ODCs of the genus *Leishmania*. Although this model has parameters that could be improved, with the available information and methods, probably this would be the closest model to *L. amazonensis* ODC that could be obtained *in silico*.

Additionally, the refined three-dimensional model was aligned with the known structure of the human ODC (2ON3), using TM-align, obtaining a normalized value with the human structure of 0.95657 (out of 1) (Appendix A). Therefore, despite the lower resolution obtained, this should not affect the validation of the drugs since they tend to bind to the active center, which does present good folding parameters and would meet the conditions for the use of docking. Thus, the structure of *L. amazonensis* ODC used in this study corresponds to iteration number 10.

#### 3.2.3. Identification and Assessment of ODC and NUO Binding to Compounds

Before docking, the ligand binding sites (frequently catalytic binding sites) in the different proteins were identified. The ligand binding site of NUO (5XTD) is well characterized in the literature. This center was first described for rotenone by Darrouzet et al. [47] at the intersection of the peripheral and membranous domains of complex I, specifically located between the NUOD/49 kDa (5XTD Chain Q) and NUOH/ND1 (5XTD Chain s) subunits. Subsequently, Bridges et al. [52] studied the crystallized structure, establishing that this site was located in the interface between the redox and proton-transfer domains. This includes the subunits ND1 (5XTD Chain s), NDUFS7 (5XTD Chain C), NDUFS2 (5XTD Chain Q), and ND3 (5XTD Chain j) (Figure 5a). Because the structure of *L. amazonensis* ODC was unknown and therefore its active site too, a structural alignment against the human ODC was performed to determine the residues involved in the putative binding site. Human ODC (2ON3) binding site was described by Dufe et al. [53] being formed by the residues Ser200, Phe238, and Tyr389 (Figure 5b). In addition, mouse studies suggest that His197 could be involved in the stabilization of reaction intermediates.

Using TM-Align structural alignment between the human and parasite structure (Appendix A), we identified that human ODC catalytic residues Ser200, Phe238, and Tyr389 correspond to Ser418, Phe456, and Tyr660 in our model of *L. amazonensis* ODC. Human His197 matches with His415 in *L. amazonensis* (Figure 6).

Once the inhibition centers were known, docking was performed using a combinatorial library of 6000 compounds derived from the lead molecule deguelin employing DockThor’s Virtual Screening mode. Due to the large size of NUO and the limitations of the server, only C, Q, j, and s subunits structure were used to perform them using a X = 20, Y = 20, and Z = 20 grid size centered in X = 220.7, Y = 167.1, and Z = 272.6. *Leishmania amazonensis*-modelled ODC dockings were performed using a X = 20, Y = 20, and Z = 20 grid size centered in X = 75, Y = 79.5, and Z = 69.

After the docking of the compounds was done (Appendix A), we removed duplicate molecules in the library (5159 final unique compounds) sharing the same SMILES. The average energy difference in the repeated compounds was 7.43 × 10^−3^ kcal/mol for the 5XTD and 1.42 × 10^−4^ kcal/mol for the *L. amazonensis* ODC. These results validate the resources used for library development and docking as well as the performance of this protocol.

Then we analyzed the energy values (in kcal/mol) for binding to human NUO and *L. amazonensis* ODC of the unique compounds derived from deguelin. The energy binding value is inversely proportional to affinity, so we chose molecules that improved (the energy binding was lower than that of deguelin) and worsened (the energy binding was higher than that of deguelin) their affinity for *Leishmania* ODC and human NUO, respectively (Figure 7). These molecules are the candidates to be better inhibitors of *Leishmania* ODC, that is, better drugs for leishmaniasis, but worse inhibitors of human NUO, that is, they will have less toxicity or side effects in humans.

Appendix A shows the 25 compounds with the best ratios. Then pharmacological properties of the first 5 compounds, whose ADME is shown in Appendix A, were also analyzed. Of all the compounds, the one with the best ratio and best properties was 12.11_14.

The deguelin-derived molecule with the best ratio was 12.11_14 whose binding energy to human NUO was −7.203 kcal/mol, 1.24-fold less than deguelin using the same docking protocol (−8.958 kcal/mol). In addition, its energy binding to *Leishmania* ODC was −10.646 kcal/mol, 1.07-fold higher than deguelin (−9.932 kcal/mol), using the same docking protocol. These differences can be explained using PLIP (Figure 8). The deguelin-derived molecule 12.11_14 only establishes 2 hydrogen bonds and 2 hydrophobic interactions with human NUO while *Leishmania* ODC establishes 2 hydrogen bonds, 8 hydrophobic interactions, and 1 π-stacking interaction.

## 4. Discussion

Leishmaniasis affects many people each year, with many of them at serious risk. In the coming years, due to climate change, human may present an increased risk of infection due to an expansion of the ecological niche of the vectors causing the disease [54]. This makes the search for new drugs very necessary to facilitate the control, mortality, and morbidity of future epidemics. Moreover, it would be important to have different drugs available to avoid possible future resistance to treatments.

Deguelin is rotenone derivative, obtained from the *cubé resin* of *Lonchocarpus utilis*. There are several studies of the inhibitory potential of deguelin and other rotenone derivatives as ODC inhibitors [55,56], with anti-leishmaniasis activity being described [57], confirming the potential of this molecule as a lead for future pharmacological development. ODC catalizes the decarboxylation of ornithine to putrescine. This is the first step of the polyamine biosynthetic pathway, which has been described as essential for the survival of the *Leishmania* genus [53,58,59,60] with the ornithine to putrescine step being the most limiting of the pathway [53]. In particular, it has been found that the transformation that takes place in the ODC has a key role in the survival and proliferation of *Leishmania* in their hosts [61,62], because the catalytic products, the polyamines, are essential for normal cell growth and differentiation [63]. Taken together, these studies reinforce our results suggesting deguelin as an ODC ligand, as well as that ODC can be a good therapeutic target [53,64], since its inhibition would cause the death of the parasite [65].

We have modeled the structure of *L. amazonensis* ODC, unknown until now, as a preliminary step to define the binding of deguelin to the protein. Although the available methods have been used to obtain the most reliable model, some regions are poorly resolved. In principle, this should not affect the docking results since the area where the inhibition center is located has good quality and a great similarity to the human protein in both structure and residues involved in the inhibitory center. The regions with poor quality are mainly those involved in the dimerization of ODCs, which is essential for its correct functioning in eukaryotes [66]. An exhaustive study of the ODC of *L. amazonensis* could identify molecules capable of preventing the formation of this homodimer, thus making possible another mechanism of inhibition. Despite this, the results obtained are extremely accurate. Owing to the advances in artificial intelligence and omics, prediction algorithms appear which can obtain results almost identical to those obtained in the traditional way, in a much faster way [67].

The bioinformatic analysis carried out also shows that deguelin is a possible ligand of human NUO and our docking results show this. As *L. amazonensis* is a parasite and coexists with the host, from this point on, the work will focus on developing compounds derived from deguelin showing increasing affinities for the parasite ODC and decreasing affinities for the *H. sapiens* NUO. Due to the large size of NUO (964.01 kDa), docking could only be performed using the active center. Despite this fact, we believe that this does not affect the results obtained, since the docking was able to resolve the ligand–protein interactions without any problem. In the future, with the improvement they are carrying out in this area, it might be possible to use the complete structure instead.

Since deguelin binds to the ODC of *Leishmania* and human NUO, we have generated a small library of deguelin derivatives, and we have analyzed its binding to the two proteins looking for analogues that increase its affinity for ODC and decrease it for NUO. There were also limitations in terms of the number of molecules studied, with 5159 single molecules being analyzed. It is possible that in the next few years studies will allow a much larger number of molecules to be identified and better therapeutic candidates to be obtained. Even with all this, the results should be taken with caution. Docking is a great advance in CADD, but it is not yet at the level of cell-based assays. That is why this work should be taken as a precursor study to in vivo research. Nevertheless, the values obtained in molecules such as 12.11_14 demonstrate the quality of deguelin as a lead, as well as the protocol used in its improvement. The binding values that it presents give us the confidence that it is possible that a deguelin derivative could become available as a treatment for leishmaniasis.

## 5. Conclusions

The integration of biological and ethnobotanical information with bioinformatic resources demonstrated that deguelin is a ligand for ODC, a protein that is a possible target for the treatment of leishmaniasis. Furthermore, in this work we also show methods to determine the structure of the *L. amazonensis* ODC (still unknown) from its sequence with a degree of refinement sufficient to perform ligand coupling analysis. The tools used in this work can be useful for the same goals with other proteins. We also show how docking analysis can be performed with very large protein complexes like human NUO using only the known structure of their active centers. Finally, we also show tools that allow the development of a small library of compounds whose properties as ligand can also be tested to gain affinity and specificity, with the consequent narrowing of chemical molecules to be tested, which implies a huge saving of resources and time. Although we are aware that these in silico results must be corroborated in future with in vitro studies, we understand that the contributions of the work can be a tool and guide that can be used by other authors and for other targets and other pathologies. It is also important to highlight the importance of research on this type of epidemic diseases mediated by parasites since climate change may affect vector distribution world-wide, provoking an increase in risk of infection/exposure.

## Figures and Tables

**Figure 1 biology-11-00133-f001:**
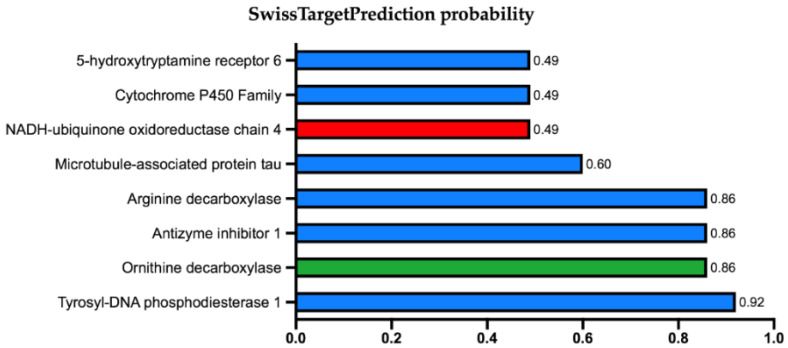
SwissTargetPrediction results for deguelin binding to human proteins. Results are shown in percentage and with cytochrome P450 family the results were grouped. In green are shown the possible therapeutic targets, in red are shown those targets with toxic effects described in the literature, and in blue are shown targets that do not show any adverse pharmacological effect in the literature.

**Figure 2 biology-11-00133-f002:**
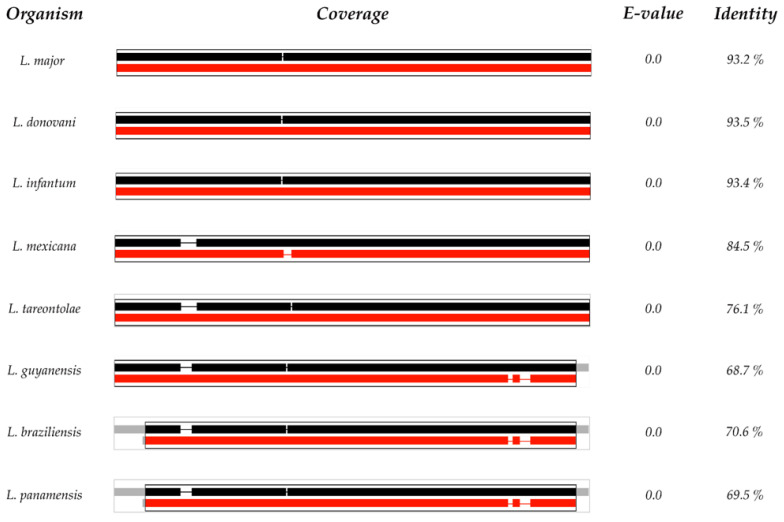
Uniprot Blast results. E-values and percentages of sequence similarity of *L. amazonensis* were compared with other *Leishmania* species. *L. amazonensis* sequence is represented in black, other *Leishmania* specie sequences are represented in red, and coverage is represented by box size.

**Figure 3 biology-11-00133-f003:**
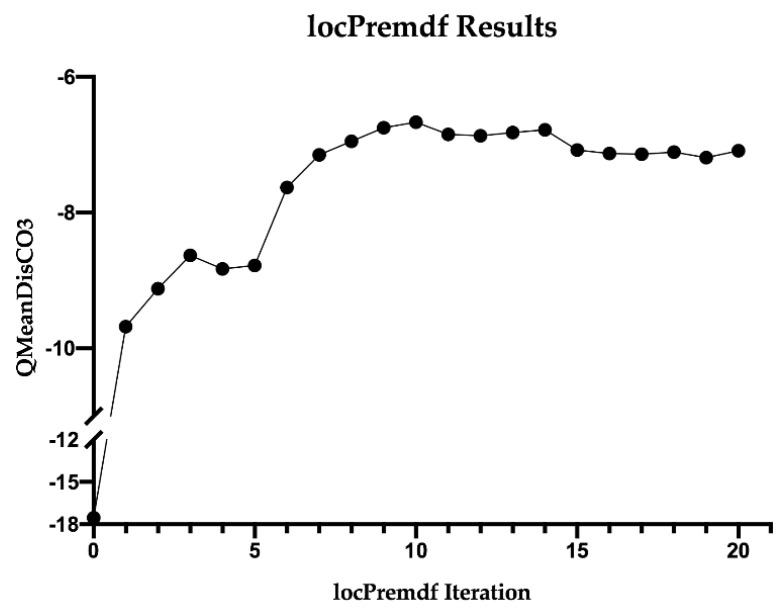
locPremdf results. Iteration 0 corresponds to initial *L. amazonensis* ODC I-Tasser modeling. QMeanDisCo3 results are scored by QMEAN4. A higher value indicates better folding.

**Figure 4 biology-11-00133-f004:**
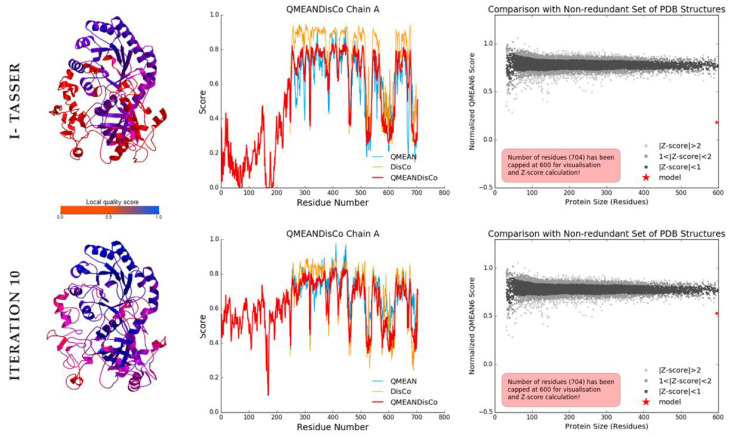
Comparison between the initial ODC model generated in I-Tasser against to iteration 10 model. Left—Evaluation of the three-dimensional model for local residues. Center—Plots by number of residues. Right—Value of the analyzed model compared to the QMeanDisCo3 validation set. Iteration number 10 was a great improvement of the model.

**Figure 5 biology-11-00133-f005:**
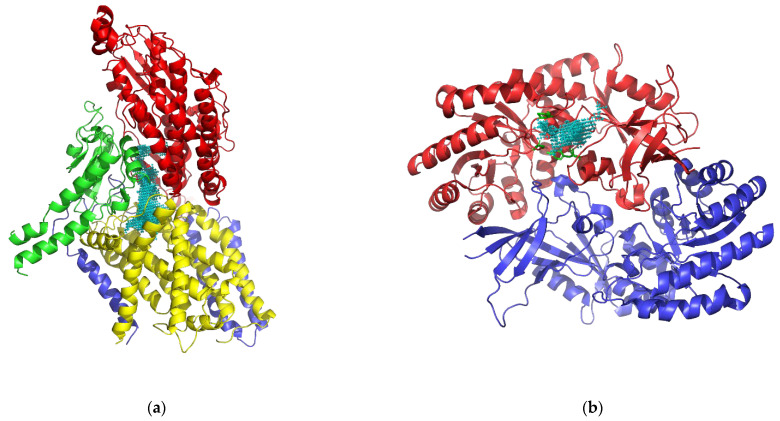
Three-dimensional structures of human NUO and ODC with their binding centers. (**a**) Human NUO (5XTD) structure. Red—Subunit Q/ NDUFS2. Green—Subunit C/ NDUFS7. Blue—Subunit j/ND3. Yellow—Subunit s/ND1. Cyan—Binding site cavity; (**b**) Human ODC (2ON3) dimeric structure. Red—Chain A. Blue—Chain B. Green—Catalytic residues (Ser200, Phe238, and Tyr389) showed in sticks. Cyan—Binding site cavity.

**Figure 6 biology-11-00133-f006:**
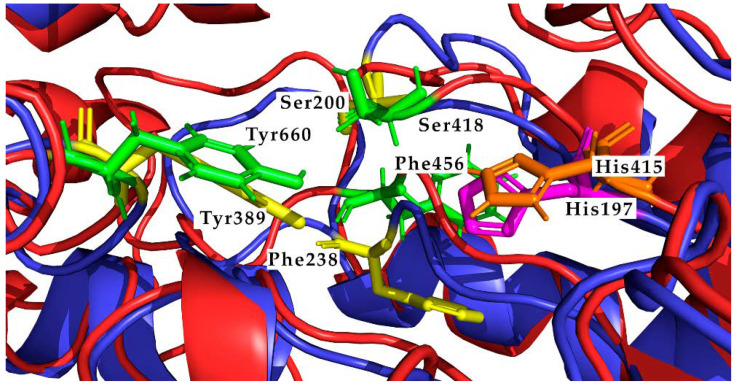
Detail of the active site structure of Human ODC (2ON3) and the active site of *L. amazonensis* ODC modeled obtained by structural alignment. Blue—Human structure (2ON3). Red—*L. amazonensis* modeled structure. Catalytic residues are shown in sticks. Yellow—Catalytic residues Ser200, Phe238 and Tyr389 determined by Dufe et al. [53] are shown in sticks. Green—Catalytic residues Ser418, Phe456, and Tyr660 determined by structural alignment. In addition, His197 (magenta) and His415 (orange) that could be involved in the stabilization of reaction intermediates have been rendered.

**Figure 7 biology-11-00133-f007:**
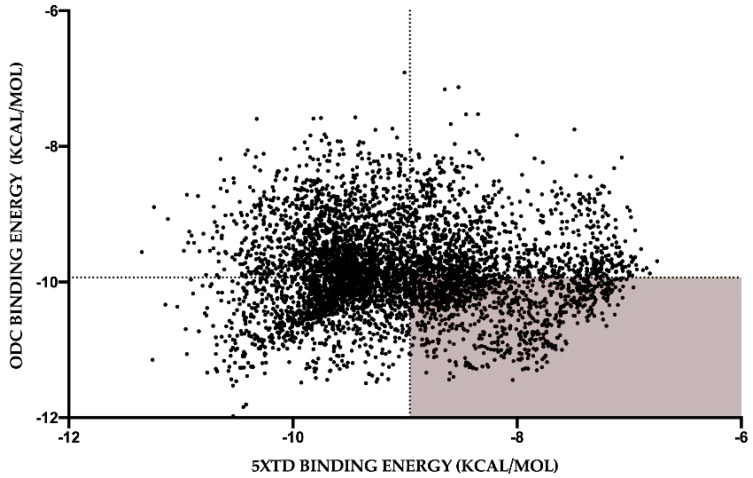
Energy binding of the 5859 cured molecules deguelin derivatives against human NUO (5XTD) and *L. amazonensis* ODC. Each point represents a single molecule. Lines represent the basal value for deguelin in both targets. In red area are highlighted those molecules with better parameters, i.e., those that improve their affinity for *Leishmania* ODC while decreasing their affinity for human NUO.

**Figure 8 biology-11-00133-f008:**
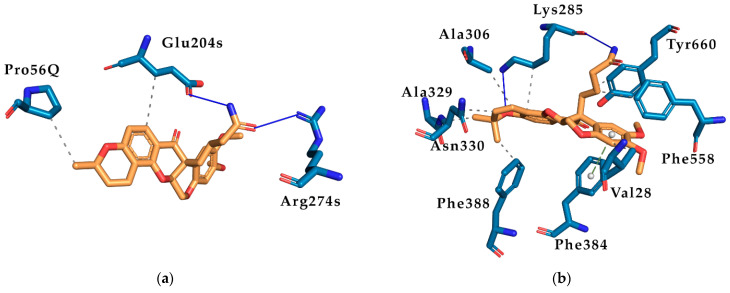
PLIP Ligand-Protein interaction of the deguelin derivative 12.11_14. (**a**) Detail of the interactions between the binding site of human NUO (5XTD) and the ligand 12.11_14; (**b**) detail of the interactions between the binding site of *Leishmania* ODC and the ligand 12.11_14. Orange—12.11_14 ligand. Dark blue—protein residues. Green—π-stacking interaction. Blue—hydrogen bonds. Grey—hydrophobic interactions.

**Table 1 biology-11-00133-t001:** Available drugs for the treatment of leishmaniasis and mechanism of action.

Drugs	Mechanism of Action
Antimonials	Antimonial complexes are administrated as Antimony (V) form which is reduced to Sb (III) in the lysosome. The reduced form inhibits some of these three targets; trypanothione, trypanothione reductase, or nucleoside topoisomerase [15].
Amphotericin B (liposomal)	Amphotericin B binds to ergosterol (sterol that is the main component of fungal and some protists cell membranes, performing the same function as cholesterol in animal cells) present in the cell membrane of parasites of the genus *Leishmania*. This binding destabilizes the membrane causing the release of that intracellular ions triggering the cell death. When administered through liposomes, the stability and specificity of amphotericin B is improved [16,17].
Miltefosine	Miltefosine is a phosphocholine analogue that inhibits the synthesis of phosphatidylcholine of the parasite, as well as the cytochrome c oxidase affecting the mitochondrial membrane potential [18].
Paromomycin	Paromomycin is an aminoglycoside antibiotic that binds to the 30S subunit of the ribosome, thus that inhibits protein synthesis [19]. Also, it has been proposed that could alter mitochondrial membrane potential and inhibition of mitochondrial respiration chain [20].
Pentamidine	Pentamidine interferes with polyamine synthesis, RNA polymerase activity, enters the protozoal cell binding to transfer RNA, and prevents the synthesis of protein, nucleic acids, phospholipids, and folate. Additionally, it is known to be an anti-inflammatory agent, xenobiotic, and an antagonist of the NMDA receptor, histone acetyltransferase, and calmodulin [21].

## Data Availability

Not applicable.

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
