# Peer review of "In Silico Research of New Therapeutics Rotenoids Derivatives against Leishmania amazonensis Infection"

_biology, 2022, doi:10.3390/biology11010133_

Round 1
Reviewer 1 Report
The authors present an in silico study to identify potential active compounds and their therapeutic targets against human leishmaniasis caused by Leishmania amazonensis. They identified deguelin has a good possible treatment as it is an ornithine decarboxylase inhibitor. In general, author suggest that CAAD analysis is a good approach that could help in drug development.
I will write some of the comments specifying the line to which they correspond.
- Line 143-148. I found these comments more appropriate for the discussion, as they are not results.
- Fig 2. Leishmania must always be written in italics.
- Line 302. I have a doubt concerning Polyamines synthesis. As the deguelin may inhibit polyamines synthesis, may it be bad for the human patient? Authors said that polyamines are necessary for normal cell function and growth (and give a quote 64), so could the inhibition of this polyamines synthesis be a problem for further use of this active compound?
- Line 313-314. Why does the author believe in this statement? Are there any other previous work or result supporting these thoughts?
- Line 345. The climate change may affect vector distribution world-wide, provoking an increase in risk of infection/exposure. Nevertheless, it does not affect to the percentage of human population susceptible to the disease.
In general, in silico studies of active molecules against different diseases is a promising tool for drug development. As the authors mention throughout the manuscript, it would be interesting to use different in vitro or in vivo assays to check the relationship of these in silico assays with the in vitro results. However, it is very interesting to perform this type of screening as a start in order to be able to narrow down the number of molecules to be evaluated.
Reviewer 2 Report
The manuscript describes a computational study to find potential inhibitors of ODC from Leishmania amazonensis. The work can be of interest to those working in computer assisted drug design and specially in antileishmanial drug discovery. However, there are several concerns that have to be taken into account.
1.- The authors performed the target selection using the software SwissTargetPrediction, from here, human ODC was selected as the potential target for deguelin, as a consequence parasite ODC was selected as the potential target of this molecule. In the same context, the NUO protein was selected for toxic effects. It is correct in principle, but, what about the potential inhibition of human ODC? Is it not important? It is desirable to give information about the potential selectivity of these molecules.
2. After all the process only one molecule is studied, according to the supplementary material, there are more compounds that supports the characteristics to be selected (higher energy for ODC that deguelin and lower energy for NUO than deguelin). At least five molecules should be characterized to obtain more information about deguelin derivatives and their potential as parasite ODC inhibitors.
3. It is recommended to include the predicted ADME-tox properties of, at least, the best five deguelin derivatives.
4. The 3D model of ODC from L. amazonensis was generated using only one program (I-tasser). Normally, it is recommendable to use more than one, to compare and be sure that the best possible model is used. What was the reason to use only one?
Round 2
Reviewer 2 Report
The manuscript was corrected according to the suggestions, therefore, I recommend its publication.